# Inactivation of Bacteriophage ɸ6 and SARS-CoV-2 in Antimicrobial Surface Tests

**DOI:** 10.3390/v15091833

**Published:** 2023-08-29

**Authors:** Sabine Poelzl, Julia Rieger, Kurt Zatloukal, Stefan Augl, Maximilian Stummer, Andreas Hinterer, Clemens Kittinger

**Affiliations:** 1Diagnostic and Research Institute of Hygiene, Microbiology and Environmental Medicine, Medical University of Graz, Neue Stiftingtalstraße 2A, 8010 Graz, Austria; sabine.poelzl@medunigraz.at; 2Diagnostic and Research Institute of Pathology, Medical University Graz, Neue Stiftingtalstraße 6, 8010 Graz, Austria; julia.rieger@medunigraz.at (J.R.); kurt.zatloukal@medunigraz.at (K.Z.); 3Department of Materials Technology, University of Applied Sciences Upper Austria, Stelzhamerstraße 23, 4600 Wels, Austria; stefan.augl@fh-wels.at; 4INOCON Technologie GmbH, Wiener Straße 3, 4800 Attnang-Puchheim, Austria; m.stummer@inocon.at (M.S.); a.hinterer@inocon.at (A.H.)

**Keywords:** bacteriophage, ɸ6, SARS-CoV-2, antimicrobial surface

## Abstract

Due to the COVID-19 pandemic, researchers have focused on new preventive measures to limit the spread of SARS-CoV-2. One promising application is the usage of antimicrobial materials on often-touched surfaces to reduce the load of infectious virus particles. Since tests with in vitro-propagated SARS-CoV-2 require biosafety level 3 (BSL-3) laboratories with limited capacities and high costs, experiments with an appropriate surrogate like the bacteriophage ɸ6 are preferred in most studies. The aim of this study was to compare ɸ6 and SARS-CoV-2 within antiviral surface tests. Different concentrations of copper coatings on polyethylene terephthalate (PET) were used to determine their neutralizing activity against ɸ6 and SARS-CoV-2. The incubation on the different specimens led to similar inactivation of both SARS-CoV-2 and ɸ6. After 24 h, no infectious virus particles were evident on any of the tested samples. Shorter incubation periods on specimens with high copper concentrations also showed a complete inactivation. In contrast, the uncoated PET foils resulted only in a negligible reduced inactivation during the one-hour incubation. The similar reduction rate for ɸ6 and SARS-CoV-2 in our experiments provide further evidence that the bacteriophage ɸ6 is an adequate model organism for SARS-CoV-2 for this type of testing.

## 1. Introduction

The SARS-CoV-2 pandemic has triggered debit on the health care system, and also on the economy worldwide [1]. The applied prevention sanctions like social distancing, wearing medical FFP2 masks, and, finally, population-wide immunization by suitable vaccines, have contributed substantially to controlling the pandemic. Nevertheless, the number of corona infections has shown, repeatedly, a locally and seasonally dependent increase [2] through the last years, which was driven by the appearance of new variants [3]. To diminish the spread of the virus particles, any additional measure is of benefit. Although virus transmissions are mainly caused by aerosols, a prospective method to contain infectious transmissions would be the usage of antimicrobial materials on often-touched surfaces, in order to reduce the load of infectious virus particles quickly. It is already documented that coronavirus particles remain infectious up to three days on stainless steel or plastic surfaces [4] and at least for 48 h on glass surfaces [5]. Therefore, researchers are focusing on developing new antiviral materials, coatings, and other novel compounds that are effective against SARS-CoV-2. Previous studies from Warnes et al., for instance, have already confirmed that copper has a biocidal effect on the human coronavirus 229E. They observed rapid structural damage (membrane damage, loss of surface spikes, breakage, etc.) as well as a nonspecific fragmentation of the entire genome by the copper ions or indirectly by the attack of reactive oxygen species (ROS) [6]. However, testing with SARS-CoV-2 needs a very complex and costly infrastructure. First, SARS-CoV-2 is a risk group 3 pathogen requiring safety measures of a Biosafety Level 3 (BSL-3) for work with the in vitro-propagated virus. This correlates with a considerable expense of time and money. Secondly, only limited BSL-3 laboratory capacities are available, which limits research on the development of antiviral measures. In contrast, the operating principles for bacteriophages are simple, as harmless to humans, since they only infect their specific bacterial host. Another benefit of operating with bacteriophages is their relatively easy method of production and rapid quantification through plaque assays [7]. Many studies have already investigated bacteriophages as a possible model for airborne viruses. For instance, N. Turgeon et al. compared five tail-less bacteriophages with two human-pathogenic viruses including the human influenza virus H1N1 and the Newcastle disease virus. Their study claimed that the influenza virus is best represented by the bacteriophage ɸ6 of the *Cystoviridae* family [8]. Furthermore, other publications have also used this bacteriophage as a surrogate for the SARS-CoV-2 virus. For instance, experiments with chlorine disinfections [9] or antimicrobial face shields [10] revealed similar efficiencies against ɸ6 and SARS-CoV-2. In the case of the antimicrobial face shields, which were composed of PET with an antimicrobial coating of benzalkonium chloride, a 100% viral inactivation of both ɸ6 and SARS-CoV-2 was detectable after just one minute of contact. This series of studies has demonstrated that this enveloped bacteriophage is a promising surrogate for SARS-CoV-2. The main reasons for this are the similarities in their structure (Figure 1), which include their lipid-enveloped nucleocapsid, their spike-proteins, and their diameter of approximately 100 nm. One important difference is the fact that ɸ6 is a double-stranded RNA phage, while the genomes of the influenza virus or coronavirus consist of single-stranded RNA [11].

The aim of the study was to investigate the antiviral activity of coatings with different copper concentrations against the model organism ɸ6 and SARS-CoV-2. These experiments provide further evidence that these two virus types show similar properties on the tested surfaces.

## 2. Materials and Methods

### 2.1. Coating Preparation

A polyethylene terephthalate (PET) foil with a dimension of 50 × 50 × 0.365 mm was used as substrate. The foils were coated with commercially available copper particles from Eckart GmbH (Hartenstein, Germany) (average diameter = 25 µm) by means of Atmospheric Pressure Plasma Deposition (APPD) using an INOCON InoCoat3 Plasma Jet (INOCON Technologie GmbH, Attnang-Puchheim, Austria).

Scanning electron microscope (SEM) images were taken to determine the amount of copper deposited on the thermoplastic surface (TESCAN MIRA3 Field Emission Scanning Electron Microscope). Moreover, the percentage of the surface covered with copper is depicted in the SEM images, but the height of the particles is not considered. The SEM images and copper loadings of the tested specimens are shown in Figure 2. All tested coatings with copper were compared to uncoated foils without copper, which were used as reference. All samples were produced aseptically and were individually packaged in a plastic film to ensure sterility while transporting them to the microbiological laboratory.

### 2.2. Antiviral Surface Tests

In general, the ISO 21702:2019 Measurement of antiviral activity on plastics and other non-porous surfaces [13] and ISO 18071:2016 Fine ceramics—Determination of antiviral activity of semiconducting photocatalytic materials under indoor lighting environment—Test method using bacteriophage Q-beta [14] were followed. The experimental setup for investigations with SARS-CoV-2 was similar that for the bacteriophage ɸ6, except for the evaluation, which required the use of different techniques. Therefore, the SARS-CoV-2 load was assessed by real-time quantitative polymerase chain reaction (RT-qPCR) instead of plaque assay. In addition, an infection assay (virus neutralization test) with VeroE6 cells after the incubation on the specimens was conducted to determine the infectivity of the SARS-CoV-2 virus particles recovered from the surfaces.

#### 2.2.1. Testing of Antiviral Activity with ɸ6

Bacteriophages and host: For the experimental setup, the bacteriophage ɸ6 DSM 21518 and its host *Pseudomonas syringae* (*P. syringae*) DSM 21482 were purchased from a collection of the Leibniz Institute DSMZ (German Collection of Microorganisms and Cell Cultures GmbH, Braunschweig, Germany).

The bacteriophage propagation was done according to the manufacturer’s specifications: *P. syringae* was cultivated overnight in lysogeny broth (LB, Carl Roth GmbH + Co Kg, Karlsruhe, Germany) containing CaCl_2_ (Merck KGaA, Darmstadt, Germany) at 25 °C and 110 rpm. The overnight host cultures were diluted at 1:50 into 10 mL fresh LB media and grown to OD_600_ 0.2. To determine the cell density by OD_600_, an UV/VIS spectrophotometer (VWR International GmbH, Vienna, Austria) was used. The culture was infected with an approximate multiplicity of infection (MOI) of 0.1 and then incubated for 4 h at 25 °C and 50 rpm. The suspension was stored at 4 °C overnight. On the next day, the phage lysate was centrifuged (10,000× *g*, 20 min, 4 °C) and then filtered through a sterile syringe 0.2 µm filter (VWR International GmbH, Austria). The phage suspension was stored at 4 °C and the titer was determined by plaque assay.

Experimental procedure: The viability of the bacteriophage particles on the copper-coated specimens were carried out according to ISO 21702:2019 [13]. The sterile surfaces were inoculated with 150 µL viral suspensions with an expected viral titer of 1–4 × 10^7^ plaque forming units (PFU)/mL and were then covered by a sterilized 40 × 40 mm PET film. Immediately after inoculation (0 h), as well as after 10 min (min), 1 h (h), and 24 h of exposure at 36 °C ± 2 °C in a humid chamber with ~96% relative humidity (RH), remaining infectious viral particles were recovered by using 10 mL SCDLP medium as neutralizer. After washing four times, dilutions in peptone saline solution (Carl Roth GmbH + Co Kg, Karlsruhe, Germany) were prepared.

Plaque assay: For the bacteriophage plaque assay, the ISO 18071:2016 [14] was applied. Therefore, 0.1 mL of the appropriate host and 1 mL of the viral dilution were added to 2 mL Top-Agar containing peptone, saline, yeast extract, CaCl_2_, and agar (Carl Roth GmbH + Co. Kg, Karlsruhe, Germany). After gently mixing, the solution was poured over LB agar (Carl Roth GmbH + Co. Kg, Karlsruhe, Germany) plates containing CaCl_2_. Each dilution was determined in duplicate. PFUs were counted after incubation for 24 h at 25 °C ± 2 °C. The concentration of infectious ɸ6 was calculated by multiplying the counted PFUs with the appropriate dilution coefficient. When no plaques were countable on the plates, the limit of the detection was set as 10 PFU, since 10 mL of the neutralization medium was used. Experiments were performed in two independent runs and, for each incubation time, triplicates (n = 6) were used to calculate mean and standard deviation for the antiviral activity of the specimens. The test validity was calculated through both 0 h triplicates with the following formula: (LOGmax − LOGmin)/LOGmean ≤ 0.2. A value below 0.2 indicated a valid test result.

Statistical analysis: The data are expressed as means ± standard errors calculated by the result of two independent experimental setups. All statistical analyses and graphical depictions were performed with GraphPad Prism 9.

#### 2.2.2. Testing of Antiviral Activity with SARS-CoV-2

Virus strains, cell culture, and propagation of SARS-CoV-2 virus: Human 2019-nCoV Isolate Wuhan strain (Product Description Ref-SKU: 026V-03883 Infectious cell culture supernatant of human 2019-nCoV Product Risk Group: RG3 ICTV Taxonomy: ssRNA(+)/Nidovirales/Coronaviridae/Coronavirinae/Betacoronavirus Virus name: Human 2019-nCoV ex China Strain: BavPat1/2020 Isolate: Germany ex China) was obtained via the European Virus Archive (EVAg). To propagate SARS-CoV-2 variants, African green monkey kidney epithelial cells (VeroE6) obtained from Biomedica (Vienna, Austria; VC-FTV6) were grown until confluence in 75 cm^2^ flasks with Minimal Essential Medium (MEM, Thermo Fisher Scientific, Waltham, MA, USA), including 5% Fetal Calf Serum (FCS, Thermo Fisher Scientific, Waltham, MA, USA), 2% L-glutamine (Merck KGaA, Darmstadt, Germany) and 1% Penicillin-Streptomycin (PenStrep, (Thermo Fisher Scientific). Cells were infected for 1 h at 37 °C and 5% CO_2_. After 1 h of infection, the cell culture medium was changed and pre-warmed. Then, serum-free Gibco OptiPRO (Thermo Fisher Scientific) was added. The flasks were incubated at the same conditions as mentioned above for a further 72–96 h, depending on the cytopathic effect (CPE) of VeroE6 cells. Prior to virus harvest, the infected cell culture was twice frozen at −80 °C and thawed to burst the cells and release the intracellular virus. To eliminate cellular debris, the suspension was first centrifuged at 3000× *g* for 10 min, then subsequently, the liquid phase was filtrated through a 0.2 µm syringe filter (Thermo Fisher Scientific). Virus stocks were stored in the BSL-3 environment at −80 °C.

To determine the virus titer, a focus forming assay was performed. VeroE6 cells were seeded in 48-well plates (Corning Incorporated, Austin, TX, USA) 24 h prior to infection at a density of 5 × 10^4^ cells/well. The virus suspension was diluted in steps of 10 with MEM with 2% FCS and cells were infected with 200 µL of virus suspensions for 1 h at 37 °C and 5% CO_2_. The infection medium was removed and 400 µL of an overlay consisting of MEM 2% FCS and 1.5% Carboxymethylcellulose (Sigma-Aldrich, Burlington, MA, USA) was added. The plates were incubated at 37 °C and 5% CO_2_ for 72 h, then the overlay was removed and cells were fixed in the wells with 4% neutral buffered formalin for immunohistochemical staining [15].

All working steps with the infectious SARS-CoV-2 virus were performed under BSL-3 conditions [16].

SARS-CoV-2 infection assay: Experiments with SARS-CoV-2 were performed essentially as described by Kicker et al. [17]. VeroE6 cells (3 × 10^4^ cells/well in MEM with 2% FCS) were seeded into 48-well plates 24 h prior to infection under BSL-2 conditions. Test specimens were sterilized with 70% Ethanol (EtOH) before they were used for the neutralization assay. The virus stock was diluted with MEM without FCS to 43 PFU/µL for infection. Each plate was pre-incubated with 150 µL of virus dilution at room temperature (RT) for 0 h and at 37 °C for 1 h and 24 h, respectively. After the pre-incubation time, the virus dilution was rinsed from the plates with 2 mL cell culture medium (MEM without FCS). From this supernatant, 140 µL were collected to determine the amount of virus used for the infection by RT-qPCR (virus input). From the remaining supernatant, 200 µL were applied to each well with VeroE6 cells that were then incubated for 60 min at 37 °C with 5% CO_2_ for infection. Thereafter, cells were washed once with MEM without FCS, and then 300 µL fresh pre-warmed cell culture medium (MEM containing 2% FCS) was added to each well. After 48 h of incubation at 37 °C and 5% CO_2_, a further 140 µL of cell culture medium was collected and RNA was isolated to determine virus copy numbers in the supernatant by RT-qPCR (virus neutralization test, t = 48). In addition, the viral suspension was incubated with medium, but without applying it to a specimen (positive control). For the negative control, medium was used without the viral suspension. Three replicates were used for each specimen (virus input, n = 3), which in turn infected three wells each (virus neutralization test, t = 48, n = 9).

Determination of virus concentration with RT-qPCR: Viral RNA was isolated from cell culture medium supernatants by using QIAamp^®^ Viral RNA Mini Kit (QIAGEN GmbH, Hilden, Germany). To detect the viral load in the samples, the RT-qPCR was performed based on the Center of Disease Control and Prevention (CDC) recommendation [18] using QuantiTect Multiplex RT-PCR Kit (QIAGEN GmbH) with a Rotor Gene Q cycler (QIAGEN GmbH) using primer pairs shown in Table 1.

Virus replication was assessed in cell culture supernatants by comparing Cycle Threshold (Ct) values at the time point of infection with Ct values after different time-periods of culturing (allowing the virus to replicate). By comparing Ct values of cells infected with viral suspension without substance (positive controls for maximal virus replication) with Ct values of cells incubated with the cooper-treated viral suspension obtained from the specimen at the end of the cultivation period (e.g., 48 h), the inhibitory effect on virus replication can be calculated. Cells cultivated without any virus were used to determine RT-qPCR background values. All Ct values higher than 40 were indicated as not detectable (nd) due to the technical limit of detection, and all values calculated and below the background of the internal controls were considered as below background (bb). To calculate viral copy numbers based on the RT-qPCR Ct values, a calibration curve based on a certified RNA standard from American Type Culture Collection (ATCC, VR-1986DTM) was used. This standard contains 4.73 × 10^3^ genome copies per 1 µL. Viral copy numbers of virus input and t = 48 were calculated using the resulting equation y = −1.51x + 38.357.

Statistical analysis: The data are expressed as means ± standard errors and all statistical analyses and graphical depictions were performed with GraphPad Prism 9.

## 3. Results

### 3.1. Effect of Cooper Coated Surfaces on Bacteriophage ɸ6

An inoculum from 1.89 × 10^7^ to 3.56 × 10^7^ PFU/mL was incubated on PET foils with and without copper (reference) for up to one day at 37 °C and at a RH of ~96%. The viral suspensions on the specimens were then harvested by a neutralizer medium and a plaque assay was performed to investigate the still-infectious viral particles (Figure 3). A complete reduction from over 5 log_10_ infectious bacteriophages was only observed for all specimens after the entire incubation time of 24 h (Appendix A). After 1 h of incubation, the remaining infectious viral particles were only detected on the specimens without copper (reference). However, even the incubation on these uncoated references revealed a decrease in the infectious viral load of approximately 0.5 log_10_ compared to the initially applied load. Based on the 10 min incubation, a different infectious virus load was detected in each sample depending on the applied copper concentration. On the specimens with the highest copper concentration (Cu2), the phage ɸ6 was not detectable. The other specimens showed an infectious viral amount between 2.33 × 10^1^ PFU/mL and 7.10 × 10^3^ PFU/mL on the surfaces. The lowest decrease can be attributed to Cu0.25, which had the least copper loading of all of the tested specimens. In contrast to the copper-coated samples, the reference had only a negligible reduction from 1.86 × 10^6^ to 1.82 × 10^6^ PFU/mL (Appendix A) after 10 min of incubation. Further, statistical analysis of the 0 h results demonstrated that the high copper concentration of Cu2 and Cu1 already lead to a significant decrease (*p* < 0.05) in the infectious viral amount compared to the uncoated reference. For the washing of the phage suspension immediately after the inoculation on the specimens, the reference showed 1.86 × 10^6^ PFU/mL, while for Cu2 and Cu1, approximately 7 × 10^5^ PFU/mL was calculated. The other specimens (Cu0.5 and Cu0.25), which contained less copper on the surface, had a similar viral particle concentration (3.12 × 10^6^ PFU/mL and 2.33 × 10^6^ PFU/mL) as the reference. Therefore, the decrease in the infectious viral amount on both of these specimens was not significant compared to the uncoated reference. These results confirmed that even brief contact with a surface with high copper concentrations could immensely reduce the amount of infectious viral particles.

### 3.2. Effect of Copper-Coated Surfaces on SARS-CoV-2

The evenly distributed SARS-CoV-2 virus stock (43 PFU/µL) on the PET foils with and without copper, which served as reference, were incubated for 0 h, 1 h, and 24 h at 37 °C and 5% CO_2_. Afterwards, the viral suspensions were rinsed with 2 mL of cell culture medium and the viral concentration was determined by RT-qPCR (virus input for virus neutralization test, Figure 4). In order to determine whether the virus particles were still infectious after the incubation on the specimen, VeroE6 cells were infected with the supernatant as described in the Section 2. After 48 h of incubation at 37 °C and 5% CO_2_, the amount of SARS-CoV-2 RNA was determined by RT-qPCR and the viral copies were calculated using an international reference standard (virus neutralization test, Figure 5).

The data of the virus input in Figure 4 demonstrated a reduction in virus RNA of over 7 log_10_ compared to positive control for all copper-coated specimens after 24 h of incubation on the surfaces. In contrast, 2.61 × 10^3^ viral particles (based on calculation of RNA copies of SARS-CoV-2) were recovered from the reference, but showed no infectivity in the virus neutralization test (Figure 5). Already after incubation for 1 h, a marked reduction in SARS-CoV-2 RNA was observed for Cu2 and Cu1 (around 5 log_10_), whereas a lower concentration of copper coating yielded similar virus RNA copies to the reference (reduction around 2 log_10_, Appendix A). In order to investigate whether the recovered RNA reflected infectious SARS-CoV-2 particles, a virus neutralization test was performed with the material recovered from the surfaces. After 1 h of incubation on the surfaces, infectious SARS-CoV-2 was only detected for the reference and the samples recovered from the Cu0.25 specimens, which yielded variable results (Appendix A and Appendix A). For these specimens, five replicates revealed a good antiviral effect with a range between 3.03 × 10^0^ and 5.06 × 10^1^ RNA copies after 48 h of incubation time, while the other four experimental replicates yielded up to 1.45 × 10^6^ RNA copies, which were comparable with the viral amount detected on the reference. On average, the samples from the surfaces coated with Cu0.5, Cu1, and Cu2 showed no SARS-CoV-2 infection, because the amount of virus particles was below the background (bb) measured for this experiment. After 24 h of incubation on the surfaces, only the reference showed infectious virus in one of the replicates. No infectious virus was detectable in any of the samples recovered from the copper-coated specimens (Figure 5).

## 4. Discussion

The pandemic of the last three years motivated many researchers to work on antimicrobial compounds, yet the facilities to test them against SARS-CoV-2 are substantially limited by access to a BSL-3 laboratory. One way to overcome this bottleneck is to use model organisms that do not require that much safety management. Until now, many bacteriophages have been described to be a good alternative to test antiviral activity under BSL-1 conditions [8,19]. However, until now, comparisons between bacteriophages and SARS-CoV-2 under comparable conditions are scarce.

We have tested four specimens with different copper loadings: Cu2 (5.6% copper loading), Cu1 (1.6% copper loading), Cu0.5 (0.7% copper loading), and Cu0.25 (0.6% copper loading). In addition, uncoated foils served as reference. The results of our investigations showed that both SARS-CoV-2 and ɸ6 revealed a complete loss of infectivity after an incubation of 24 h on all tested specimens. Even after 1 h, high copper concentrations (specimen Cu2, Cu1, and Cu0.5) led to an approximately 5 log_10_ reduction in ɸ6 and an over 6 log_10_ reduction in SARS-CoV-2 infectivity. For the lowest copper concentration of Cu0.25, a difference in the antiviral activity was detectable between the two organisms, but this was due to the elevated standard deviation within the neutralization tests of SARS-CoV-2 (Appendix A). Thus, some data from the Cu0.25 results also showed a consensus between ɸ6 and SARS-CoV-2.

The additionally implemented short incubation time of 10 min with ɸ6 also achieved a significant reduction in all copper-coated specimens compared to the reference. These 10 min results revealed a dependence of the antiviral effect on the used copper loading. A higher amount of copper in the coatings caused a faster elimination of the phage ɸ6 on the surfaces. However, these short-term incubation experiments with SARS-CoV-2 were not possible, due to the different and time-intensive handling in the BSL-3 facility. Nevertheless, the data from the other incubation periods showed very similar results between ɸ6 and SARS-CoV-2.

Interestingly, a reduction in both virus particles was also observed after 24 h on the reference material PET. This effect of an antimicrobial activity by PET surfaces on both the bacteria and bacteriophages has been observed a few times before in different setups in our laboratory. This effect might be due to different concentrations of plasticizers in the polyethylene terephthalate foils, which may have a neutralization effect on microorganisms and are not specified by the manufactures. As described by Warnes et al., virus particles from HuCoV-229E can show different survival rates on the tested surfaces [6]. Although an antiviral activity of PET against both SARS-CoV-2 and ɸ6 was evident in our research (only after 24 h of incubation), we want to highlight the fast efficiency of the copper coatings. This rapid inactivation of the virus particles on frequently touched surfaces by coatings could be a helpful contribution to the horizontal approach to minimizing the risk of subsequent infections. In addition to the observed effectivity of the coatings used within this study, we also demonstrate the comparability of the results obtained with the two different virus types that were tested.

In our studies, the bacteriophage ɸ6 may serve as a substitute for SARS-CoV-2 and thus allows the opportunity for tests with, e.g., infectious droplets (aerosols), different exposure conditions (e.g., proteins), and durability examinations of the surfaces, which can hardly be performed with SARS-CoV-2 under BSL-3 conditions. In summary, the greatly reduced safety concepts prescribed for work with bacteriophages allow access to extensive testing of several antiviral agents and disinfection conditions that would not be feasible with SARS-CoV-2.

## Figures and Tables

**Figure 1 viruses-15-01833-f001:**
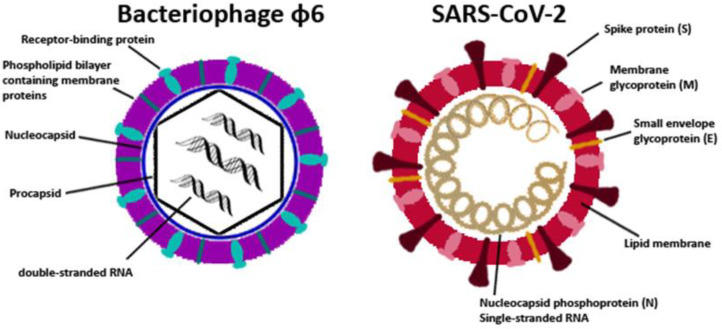
Structure of bacteriophage ɸ6 and SARS-CoV-2. Adapted from Laurinavičius et al. [12] and N. Chams et al. [1]. Created by S. Poelzl with Adobe Photoshop CS2 Version 9.0.

**Figure 2 viruses-15-01833-f002:**
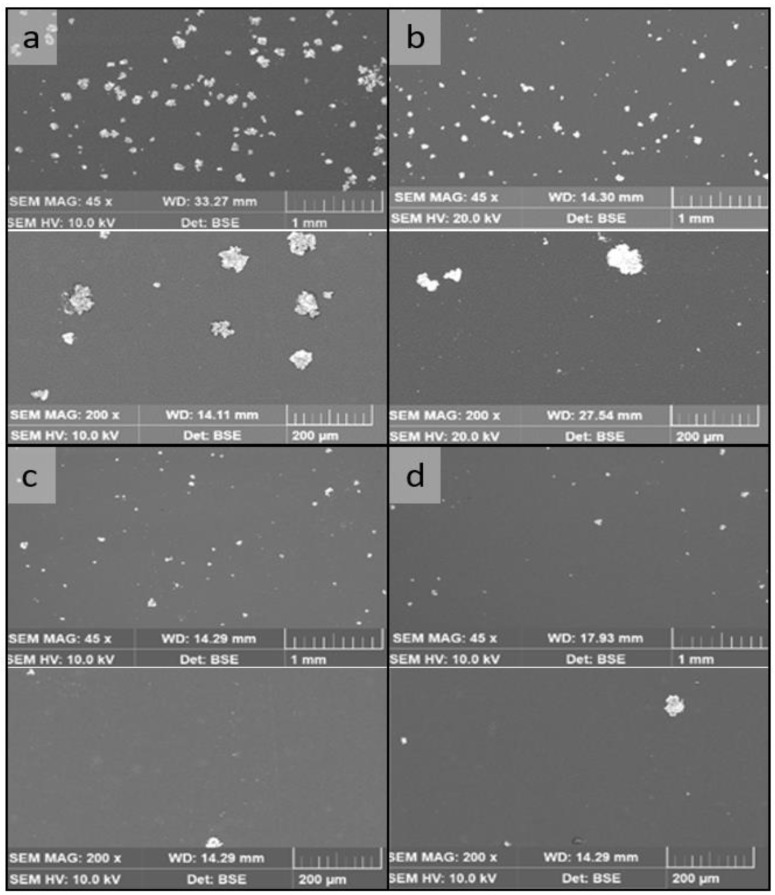
Two scanning electron microscope images (SEM) were taken of each sample, with an overview image (top, 1 mm) and a detail image (bottom, 200 µm) to visualize the different copper loadings on the samples: (**a**) Cu2 (5.6% copper loading); (**b**) Cu1 (1.6% copper loading); (**c**) Cu0.5 (0.7% copper loading); (**d**) Cu0.25 (0.6% copper loading).

**Figure 3 viruses-15-01833-f003:**
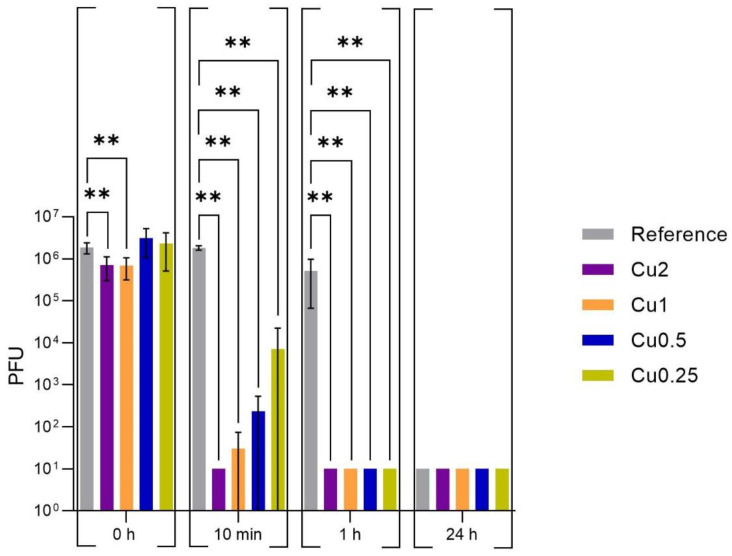
Quantification of infectious bacteriophages ɸ6 on different copper-coated specimens. The surfaces were inoculated with a viral load of approximately 10^7^ PFU/mL. The different temporal incubations of the samples were performed at 37 °C and at a RH of ~96%, before the bacteriophages were harvested and checked for their infectivity by plaque assay. Further, 0 h shows the recovered infectious phage particles immediately after inoculation on the tested specimens. The times of 10 min, 1 h, and 24 h reflect the recovery of the infectious viral amount after the different incubation times on the samples. An uncoated specimen served as reference. The error bars indicate the standard errors of the respective means, which were composed of triplicates in two independent runs (n = 6). The limit of detection was set as 10 PFU. Statistically significant differences between the uncoated reference and the copper-coated surfaces within the same incubation time are marked (mean with 95% Cl; Mann–Whitney U test; *p*-value: <0.05; ** indicates statistical significance with a *p*-value below 0.02).

**Figure 4 viruses-15-01833-f004:**
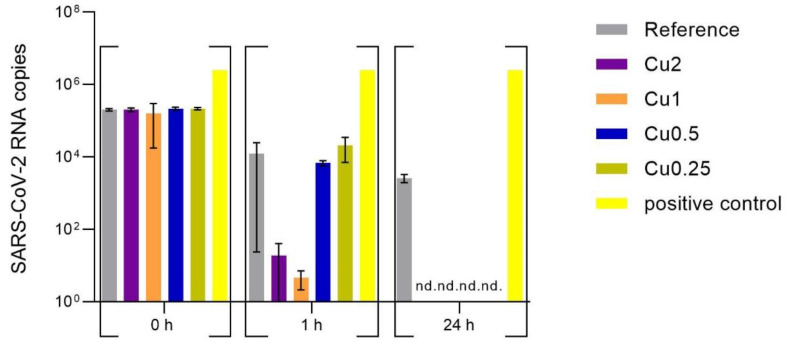
Quantification of SARS-CoV-2 RNA recovered from the tested specimens and used as input for the virus neutralization test by RT-qPCR. The surfaces of copper-coated and uncoated (reference) specimens were covered with a serum-free SARS-CoV-2 virus stock (43 PFU/µL). The incubation of the inoculated specimens for 1 h and 24 h was performed at 37 °C and 5% CO_2_. After the incubation periods, the virus suspensions were rinsed from the surfaces with 2 mL medium. The washing step was also performed immediately after the application to achieve the initial concentration (0 h). For the positive control, the viral suspension was incubated with medium but without applying a specimen. The virus copies were determined by RT-qPCR (nd, no virus RNA detected). The error bars indicate the standard errors of the respective means, which were composed of triplicates (n = 3). Statistically, there is no significant difference between the virus input from the treated surfaces and the reference, as the scatter of the individual values is too large.

**Figure 5 viruses-15-01833-f005:**
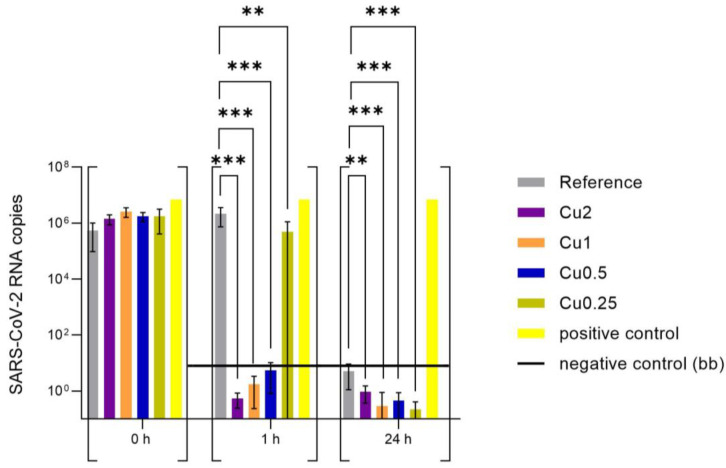
Infectivity of viral SARS-CoV-2 particles recovered from the tested specimens (virus neutralization test). After 0 h (rinsed from specimens immediately after application), 1 h, and 24 h of pre-inoculation on the specimens, VeroE6 cells were infected with the viral suspension rinsed from the surfaces. After 48 h of incubation at 37 °C and 5% CO_2_, viral RNA in the supernatant was isolated and virus copies were determined with RT-qPCR (t = 48). For each specimen, triplicates were tested, which in turn infected three wells (n = 9). For the positive control, the viral suspension was incubated with medium but without applying a specimen. No virus was added for the negative control that served to determine the background of the experiment. All values below were considered as below background (bb). Statistically significant differences between the uncoated reference and the copper-coated surfaces within the same incubation time are marked (mean with 95% Cl; Mann–Whitney U test; *p*-value: <0.05; ** indicates statistical significance with a *p*-value below 0.02; *** indicates statistical significance with a *p*-value below 0.001).

**Table 1 viruses-15-01833-t001:** Primer pairs and probe used for the RT-qPCR.

2019-nCoV_N2-F 2019-nCoV_N2 Forward Primer	5′-TTA CAA ACA TTG GCC GCA AA-3′
2019-nCoV_N2-R 2019-nCoV_N2 Reverse Primer	5′-GCG CGA CAT TCC GAA GAA-3′
2019-nCoV_N2-P 2019-nCoV_N2 Probe	5′-FAM-ACA ATT TGC/ZEN/CCC CAG CGC TTC AG-3IABkFQ-3′ FAM, BHQ-1

## Data Availability

The datasets generated during and/or analysed during the current study are available from the corresponding author on reasonable request.

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
