# Peer review of "Inactivation of Bacteriophage ɸ6 and SARS-CoV-2 in Antimicrobial Surface Tests"

_viruses, 2023, doi:10.3390/v15091833_

Round 1
Reviewer 1 Report
The manuscript contains quite a useful information about thorough comparative inactivation pair testing of bacteriophage Phi6 and SARS-CoV2 coronavirus on copper-coating PET. This comparative test demonstrated practicalle idenical results which showed that non-dangerous Phi6 bacteriophage may be used instead of usage of very dangerous coronavirus in experimental testing of different surfaces for inactivating properties. Phi6 bacteriophage has a lipid envelope with inserted proteins and its RNA is packed inside this envelope which is very similar to coronavirus virion structure. The only significant difference is the dsRNA inside Phi6 virion instead of ssRNA inside the SARS-CoV2 virion. The authors presented detailed introduction in which they mentioned previous usage of Phi6 by other authors as a model for another human viruses in inactivation tests which made much easier the evaluation of the inactivating properties in their experiments. In my opinion, the current manuscript is very good and useful and deserves the publication.
Author Response
Dear Reviewer 1,
thank you for the revision and the positiv feedback.
Reviewer 2 Report
The authors report the results of inactivation of SARS-CoV-2 virus on PET surfaces coated with different concentration of copper. Along they used Phi6 phage, and the level of inactivation shows a strong correlation for both viruses – this is an important point as the handling of this model organism requires less exhaustive safety measures, and thus potentially a broader range of possible inhibitory substances could be tested and several parameters explored and orientationally set. Time dependency of the inactivation on coated surface is also shown. One parameter the authors should comment on is the initial virus load: how does the efficiency of inactivation correlate with the initial load?
In principle, this contribution is valuable in supporting pandemic preparedness. Technically, certain Figures should be prepared better (please see the comments below). Further, Figure 5 is not cited in the text and it seems that Figure 4 sometimes is referred to as Figure 3 (please see comments below). Unfortunately, the supplementary materials were not available, and I cannot assess the results cited within.
The language of the article (especially certain passages in Materials and Methods) is too colloquial and this should be corrected. Please find below the list of remarks which I hope will find helpful.
Line 23: on any (or in any) of the tested samples
Line 23 (and throughout the text): plural of specimen is specimens
Figure 2. The labels of the figures are not well legible (even upon magnification)
Line 127: 10,000g: please use comma as a decimal separator only
Line 144: gently mixing
Line 164: was obtained
Line 180: prior to infection
Line 183: MEM with
Line 186: immunohistochemical staining
Line 191: MEM with 2% FCS
Line 194: MEM without FCS
Line 200 (and throughout the text): CO2, 2 in subscript
Lines 215-222: please organize the list of primers and probes (using semicolons, table or similar)
Line 227: PCT, please explain the abbreviation
Line 251: 10-min-incubation
Line 251: in each sample
Line 256: specimens
Line 282: copper
Line 287: do you mean Figure 4?
Line 296: Please can the authors explain what exactly is presented in Figure 4 and Figure 5. I assume Figure 4 is RT-PCR from “virus only”, and Figure 5 RT-PCR from the supernatants of infected cells. Figure 5 is not cited in the text.
Figure 4: measures of significance should be included.
Line 384: For a final strong statement, a better expression would be: “allow access to extensive testing of several antiviral agents and disinfection conditions”, or similar
Lines 386-395: Supplementary materials were not accessible along with the article to review
Line 432: the format of reference 5 is unusual
Line 463: the format of reference 17 is unusual
Author Response
Dear Reviewer 2,
thank you for the detailed revision.
Please find below in the word document the response to you comments.

Round 2
Reviewer 2 Report
The authors have improved the data presentation and the description of methods. With this, I can recommend the manuscript for publication.